

# A review of the opportunities to support pollinator populations in South African cities

Peta Brom[1], Les G. Underhill[1] and Kevin Winter[2]

[1] Department of Biological Sciences, University of Cape Town, Cape Town, Western Cape, South Africa
[2] Environmental and Geographical Sciences, University of Cape Town, Cape Town, Western Cape, South Africa

## ABSTRACT

Globally insects are declining, but some guilds of pollinators are finding refuge in urban landscapes. The body of knowledge on urban pollinators is relatively mature, which means it is now possible to begin to make generalization. Unfortunately, studies do not represent climatic regions evenly and there is a gap in research from the African continent. This study aimed to address some of the gaps on urban pollination knowledge in South Africa and to identify opportunities to improve urban habitats for pollinators. We reviewed the international literature on urban pollinators and the South African literature on pollinators with a landscape ecology focus, drawing on literature with an emphasis on agricultural and ecosystem services. The findings show that some taxa (*e.g.* large-bodied, cavity nesting bees) will exploit urban environments increasing in abundance with urban intensity. Moderately sensitive taxa (such as small-bodied, ground-nesting bees) take advantage of urban environments only if local habitats are supportive of their needs for resource provision and habitat connectivity. The South African urban poor rely on pollination services for subsistence agriculture and the reproduction of wild-foraged medicines and food. Potential interventions to improve habitat quality include strategic mowing practices, conversion of turf-grass to floral rich habitats, scientific confirmation of lists of highly attractive flowers, and inclusion of small-scale flower patches throughout the urban matrix. Further research is needed to fill the Africa gap for both specialized and generalized pollinators (Diptera, Halictids, Lepidoptera and Hopliini) in urban areas where ornamental and indigenous flowering plants are valued.

## INTRODUCTION

Overall insect populations are in decline (*Cardoso et al., 2020*; *Wagner et al., 2021*). In the last 20 years, insect biomass in Germany decreased by 75% (*Hallmann et al., 2017*) and a widespread study in the UK measured decreases in a third of wild species of pollinators between 1980 and 2013 (*Powney et al., 2019*). Loss of habitat, agrochemical-stress, fragmentation, disease, competition from aliens, and climate change are listed as the nexus

Corresponding author
Peta Brom, brompeta@gmail.com

of interlinking threats to biodiversity and insect populations alike (*Cardoso et al., 2020*; *Raven & Wagner, 2021*), and researchers are looking for ways to mitigate against losses and protect against decreases (*Samways et al., 2020*). One area of research which provides some positive potential outcomes for the fate of pollinators (at least for some taxa), is in the refuge offered by cities. Cities have variously been touted as "refuges for pollinators", but "not a panacea for all insects" (*Ives et al., 2015*; *Hall et al., 2017*; *Theodorou et al., 2020*), due to the mixed results rendered for different taxa and guilds (*Wenzel et al., 2020*). Furthermore, the global increase in the proportion of land cultivated with pollinator-dependent crops, implies a greater reliance on managed pollinators (*Aizen, Aguiar & Biesmeijer, 2019*). The potential of urban environments to contribute to the conservation and management of pollinator habitats has therefore been highlighted as an important component in conservation planning (*Hall et al., 2017*). Investigating management interventions to support them in urban landscapes therefore becomes paramount.

The body of knowledge on urban pollinators is relatively mature in the Global North and growing in the Global South, making it possible to systematically review the literature and begin to draw generalizations and comparisons (*Wenzel et al., 2020*). Unfortunately, regions are not evenly represented. In 2013, only 4% of data on pollination was produced from the entire continent of Africa (*Archer et al., 2014*). In consideration of ecosystem service and green infrastructure more generally, *du Toit et al. (2018)* found only 38% of sub-Saharan countries had carried out any research on the topic (*du Toit et al., 2018*). More recently, *Wenzel et al. (2020)* reviewed 141 studies of urban pollinators and found that most studies had been produced by temperate and developed countries (117 and 120 studies respectively) and tropical and developing countries have produced only a fraction of these (24 and 21). There is poor representation from Eastern Europe and the African continent, where they identified only a single study from Ghana (*Guenat et al., 2019*). To this, two studies on sunbirds can be added (*Coetzee, Barnard & Pauw, 2018*; *Pauw & Louw, 2012*) and one on butterflies (*Avuletey & Niba, 2014*) from South Africa. Due to regional differences, this lack of knowledge from the African continent leaves a gap in the understanding of how pollinators are responding to urban landscapes, particularly in the rapidly urbanizing context of the Global South (*Wenzel et al., 2020*).

The above gap makes it challenging to review potential opportunities for urban pollinator management in southern Africa. While developments in urban pollination studies have been nearly non-existent, there is a greater body of knowledge available from another landscape which is subject to anthropogenic change: Agriculture (*Melin et al., 2014*, *2018*). Urban landscapes are one of several examples of near-total habitat transformation, in which the natural habitat is almost completely replaced. In the urban context, the replacement is by buildings, roads, and introduced plants. Similarly, agricultural landscapes have replaced natural habitats. The latter are destroyed and usually replaced by monocultures; the most extreme of which being cereal crops, which are wind pollinated and require no animal agents for reproduction. Like urban landscapes, agriculture produces fragmentation in the natural landscape, and generates barriers to

sensitive faunal species movement through the introduction of pesticides and agro-chemicals (*Lenhardt et al., 2013*). In comparison, barriers and extirpations are generated by road networks and buildings in urban landscapes. Another similarity between the transformation of urban and agricultural landscapes is the scale and intensity at which it occurs (*Wellmann et al., 2018*). Therefore, in the absence of regional studies of pollinators in urban landscapes, studies of pollinators in agricultural landscapes become useful for reflection on the impact of land cover transformation on pollination (*Donaldson et al., 2002*; *Geslin et al., 2016*) and the potential for mitigation and supplementation. This is not to claim that findings are transferable because agricultural and urban landscapes favour different taxonomic groups, but rather that insights can be drawn by interrogating the broad patterns.

This review explores the potential role that urban landscapes can play in supporting pollinators in southern Africa. In the absence of a body of knowledge on urban pollinators in Africa, the topic is discussed by synthesizing what is known about pollinators in landscapes transformed by agriculture with studies discussing eco-system services in urban environments, and comparing the patterns observed in the region with what is known from global urban pollinator studies. Key themes are highlighted and discussed to identify areas for future research and opportunities for pollinator conservation which can inform policy. We discuss the ways social and plant patterns may be driving species assemblages and generating a demand for pollination services in South African cities.

## SURVEY METHOD

An initial literature search was conducted using the search term: "TITLE-ABS-KEY (Urban AND pollinators AND NOT pesticides AND NOT insecticides)" in Scopus in order to take in the international body of knowledge on urban pollinators. Titles and abstracts were read. Studies which did not discuss urban landscape effects on pollinators were excluded. Papers which focused on taxonomic, phylogenetic, invasion biology or evolutionary traits, and mangroves were beyond the scope of this review. Six classes of paper were selected for thorough analysis: reviews, multi-city studies, multi-region studies, those papers grouped together by taxon, but for which contradictory responses to the urban gradient were recorded, longitudinal studies, and those with novel approaches or findings. The key findings of the remaining papers were extracted and tabulated and coded for positive, negative or mixed responses to urbanization (Supplemental Data 1). This supplementary data represents a duplication of the work undertaken by *Wenzel et al. (2020)* however we aim to investigate the implications for the Global South generally, and Sub Saharan Africa more specifically.

The initial search had produced just two urban pollination studies from South Africa (*Coetzee, Barnard & Pauw, 2018*; *Pauw & Louw, 2012*). Additional searches for the South African body of knowledge with keywords did not produce satisfactory results, so instead prominent researcher's publication lists were consulted. Their co-authors and authors from their reference lists were similarly searched. Key informants were approached to suggest additional authors and studies that may be of interest. Papers were selected for in-depth reading based on their focus on landscape ecology and pollination. The results are
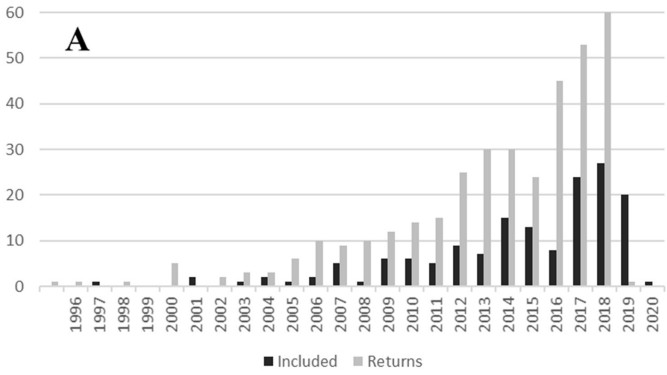
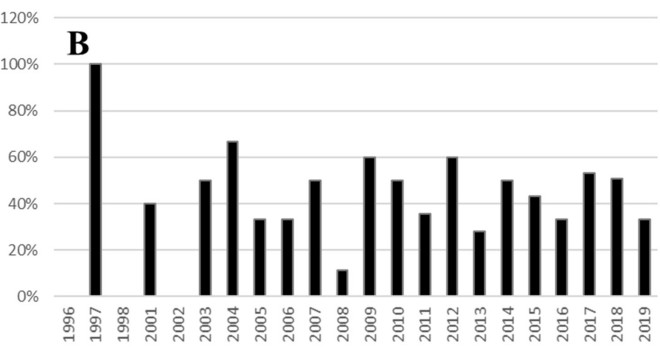

**Figure 1 Temporal spread of urban pollination studies.** (A) Number of studies meeting search criteria against number of studies selected for inclusion in the review by year. (B) Percentage of studies meeting the scope of the review by year.

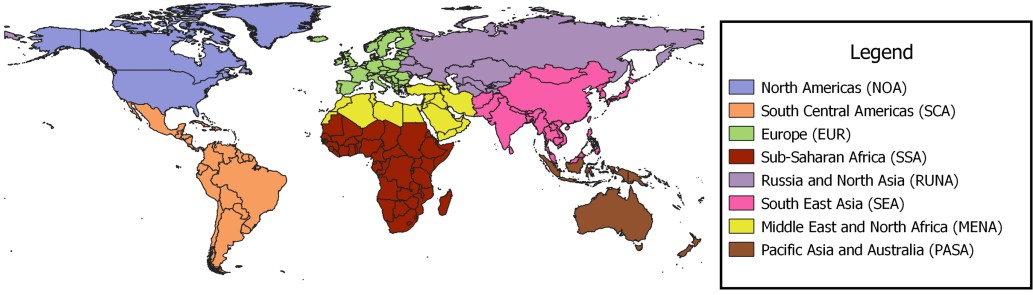

**Figure 2 Regional borders used for grouping papers on urban pollination.**

organized and discussed according to the Mobile Agent Based Ecosystem Services framework (*Kremen et al., 2007*).

# RESULTS

The initial search generated 356 documents. From those, 158 were within the scope of the study. A further 37 papers met the criteria for inclusion from the focused search on South African studies (Supplemental Data 2). The earliest paper was published in 1996, and the earliest within the scope in 1997. Publication rate grew exponentially from the start (Fig. 1A). In 2019, 60 papers met the search criteria. The percentage of relevant studies (Fig. 1B) was between 25% and 60% across all years.

## Spatial distribution

The regional borders used to define and group studies are presented in Fig. 2. Europe (EUR) and North America (NOA) combined accounted for 77% of the studies at 63 and 53 studies respectively; South, Central Americas (SCA) contributed 14 studies and Russia and Northern Asia (RUNA), Middle East and North Africa (MENA), Sub-Saharan Africa (SSA) and South East Asia (SEA) produced 1–9 studies each. Of the studies from SSA, four of the five, were from South Africa (See Table 1).

**Table 1 Global distribution of studies on urban pollinators.** More than two thirds of studies reported mixed findings or found that variables other than urbanization better explained the patterns observed in cities.

| Region | Number of papers | % of total | Positive | Negative | Neutral/Mixed |
|--------|------------------|------------|----------|----------|---------------|
| EUR | 63 | 42.0 | 12 | 6 | 46 |
| NOA | 53 | 35.3 | 4 | 10 | 39 |
| SCA | 14 | 9.3 | 3 | 1 | 11 |
| PASA | 9 | 6.0 | 0 | 1 | 8 |
| SSA | 5 | 3.3 | 0 | 2 | 3 |
| SEA | 4 | 2.7 | 0 | 0 | 4 |
| RUNA | 1 | 0.7 | 0 | 0 | 1 |
| MENA | 1 | 0.7 | 0 | 1 | 0 |
| TOTAL | 150 | 100 | 19 | 21 | 112 |

## Species representation

Of the 150 studies 105 included *Hymenoptera*, 18 *Lepidoptera*, 16 *Diptera*, eight Avifauna, five Mammalia and five *Coleoptera*. A total of 20 considered aspects relating to plant diversity, pollen trails or traits.

Whilst studies from temperate climates were dominated by *Hymenoptera*, those that focused on forest fragments, presented more varied pollinator mutualisms including nine studies on the effects of urbanisation on *Ficus* and the associated fig-wasp species, five on hummingbirds in South America (compared with two on sunbirds in South Africa and one on hummingbirds in North America), four studies on bats, and one compared flying foxes in Japan and Taiwan.

## Response to urban gradients

Studies that rendered explicitly positive or negative associations with urban intensity were more likely to include a limited number of taxon or guilds, whilst those that reported mixed findings compared orders, or categorized groups according to functional response such as nesting strategy, or body size. Neutral findings reported that variables other than urban intensity better explained the observed community structure patterns. A total of 33 studies included some or all of the explanatory variables: amount of green space in the surrounding neighbourhood, patch size, and connectivity as positively affecting pollinator diversity or as a recommendation for urban pollinator conservation.

## DISCUSSION

### A conceptual model for analyzing the interactions of mobile agents (Pollinators)

Several studies have highlighted the landscape and habitat drivers of urban pollinator communities which affect them across different scales (*Baldock, 2020*; *Bennett & Lovell, 2019*; *Plascencia & Philpott, 2017*; *Wenzel et al., 2020*). They include local habitat structure, floral abundance and distribution, urban intensity, and land-use. However, few studies attempted to place these components into a framework which gives an overview of the way
they interact with pollinators and each other in cities. *Aronson et al. (2016)* discussed urban drivers of community assemblage from the perspective of hierarchical filtering theory, which demonstrates how landscape factors filter out species from the regional pool. However, the hierarchical filtering model falls short in that it does not account for upward flows, in which local interventions can positively affect regional abundances; nor does it account for the ways in which connectivity between patches promotes mobility through otherwise hostile environments (*Fuller et al., 2008*; *Zuckerberg et al., 2011*).

In contrast, *Kremen et al. (2007)* produced a conceptual model of mobile agent-based ecosystem services (MABES). This shows the flow of influences and interactions which are interchangeable between different landscape types and ecosystem services. The premise of the model is that mobile agents (e.g. pollinators), are often segregated spatially or temporally from the location where an ecosystem service is provided. The system does not assume a unidirectional flow between all components and includes a feedback loop in the form of policy and economic intervention. In this case, it is applied to the ecosystem service of pollination; however, the model can be generalized to all mobile agents of ecosystem provision (*Kremen et al., 2007*). It is therefore appropriate to use for identifying the components of the system which are influencing pollinator assemblages across urban and agricultural landscapes.

Pollinator efficacy is predicated on the distribution of resources at landscape scale and the foraging and dispersal movements of the pollinator. MABES proposes several levels of interaction between pollinators, plants, and the environment. Land use, management, and disturbance cycles (*e.g.* for urban landscape, change in private land ownership) shape the landscape structure and the biotic and abiotic environment, and influence both the plant and pollinator communities which are interacting with each other (Fig. 3). The output is the pollination service required to produce crops and perpetuate local plant populations. How this service is valued by people impacts landscape management policy, which in turn shapes the landscape structure in a feedback loop (*Kremen et al., 2007*).

Humans therefore act as facilitation agents through their value-assessments and objective setting processes. *Kremen et al. (2007)* conceptualized the assessed value of an ecosystem service as influencing policy, but value-driven feedback loops at local scales can also inform both design and management decisions (*Freeman et al., 2012*; *Goodness, 2018*). For example, in a garden setting the desired ecosystem services may include aesthetics, psychological well-being, thermal regulation (shade), social connection (gathering space), and leisure activities (*Nahlik et al., 2012*). Depending on the ecosystem service objectives of the garden owner, different design decisions and management practices may be implemented. This feedback and acknowledgement of humans as facilitation agents in objective setting for ecosystems services is fundamentally different to the passive and singular directional model of hierarchical filtering (*Aronson et al., 2016*).

This review is structured according to the components identified by the MABES conceptual model (*Kremen et al., 2007*). In the sections below, each of the components of landscape structure, pollinator assemblage, plant assemblage, local habitat, and humans as agents of ecosystem service facilitation are considered. For each component, what is

**Geographic Context**

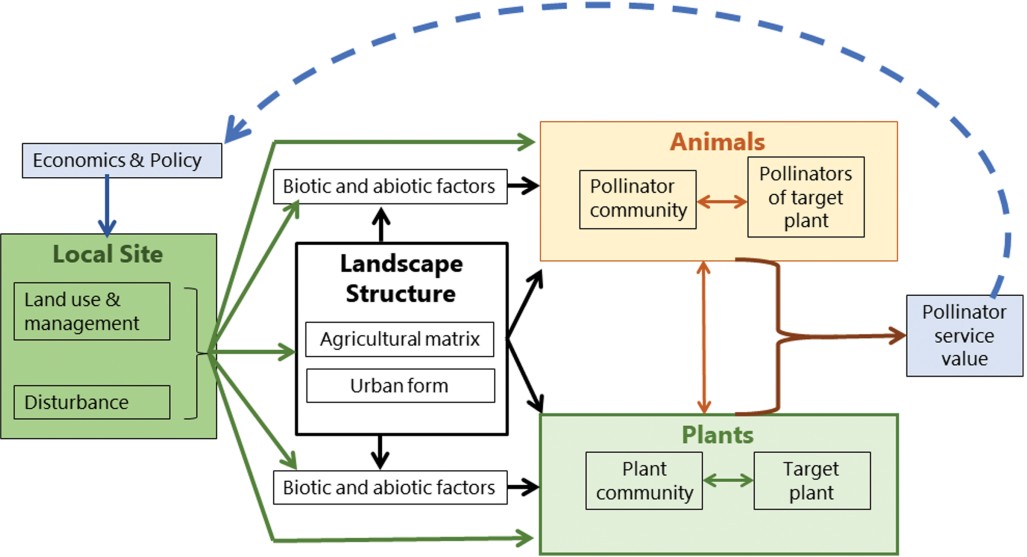

**Figure 3 Mobile agent-based ecosystem services.** A conceptual framework (*Kremen et al., 2007*) showing the drivers of pollinator community assembly.

known about urban pollinators from the international research pool is compared with the agricultural and ecosystem services studies on pollination in Africa. The urban study recommendations are then identified from the combined literature and tabulated as pollinator management opportunities. The review concludes by summarizing the research gaps and the ways in which cities might provide more hospitable landscapes and refuges for pollinators in South Africa.

## Landscape structure and spatial patterns in South African cities

South African cities are characterized by extreme social inequality and rapid urban expansion. The national Gini co-efficient is 0.63 ranking it as the most unequal society in the world (*World Bank, 2021*). The spatial arrangement of this inequality is distributed into discrete neighbourhoods stratified by the legacy of Apartheid city planning which separated urban development along racial and economic lines until 1990. African cities are expanding at a rate of 4% *per annum*, which is c. 10 times the rate of the Global North. The cause of this is expansion is predominantly due to rural to urban migration, a process through which arriving citizens tend to settle in informal structures in unplanned settlements at the urban fringe (*Myers, 2021*; *Pauleit et al., 2018*). The distribution of plants along urban, social and environmental gradients, shape the quality of local habitats and dictates where the need for pollination services will be required for meeting both provisioning and biodiversity needs (*Aronson et al. 2016*).

 Potchefstroom is a medium sized city in South Africa. *Lubbe, Siebert & Cilliers (2010)* noted an increase in native species at the lower end of the socio-economic gradient along with greater reliance on native plants for foraged food and medicine. In the larger metropolis of Cape Town, *Anderson et al. (2020)* noted an increase in biodiversity along

with wealth. The increase in diversity associated with wealth, is a phenomenon known as the "luxury effect"—a concept established through empirical evidence gathered in the Global North. Due to the rapid expansion of informal settlements at the urban periphery and the juxtaposition of new (poor) settlements infringing on remnant habitats, poor communities of the Global South may correlate with greater biodiversity by juxtaposition with adjacent wilderness or undeveloped areas (*Anderson et al., 2021*). For communities abutting or invading nature reserves, this phenomenon does not necessarily result in better access to green infrastructure or greater biodiversity within poor communities. Instead, the extreme density and lack of planning has resulted in a deficit of quality green infrastructure to the South African poor (*Venter et al., 2020*), yet urban agriculture is prevalent in cities of the Global South, with an increasing likelihood of public open space hosting animal grazing or subsistence farming at the lower end of the social economic gradient where available space exists (*Myers, 2021*). This is in addition to a shift towards gardening for provisioning ecosystem services. On the other end of the spectrum, the increase in biodiversity associated with wealth in South African cities, is also correlated with an increase in exotic species—a phenomenon that is attributed to the colonial legacies of urban gardening practices and a shift to aesthetic and cultural ES (as a status symbol) away from provisioning ES (*Lubbe, Siebert & Cilliers, 2010*; *Myers, 2021*; *Pauleit et al., 2018*). This means that there is greater demand for pollination services at the poorer end of the economic gradient and a tendency towards artificial landscapes with wealth, which likely has mixed effects on pollinator resource provision in wealthy neighbourhoods (*Leong, Kremen & Roderick, 2014*).

## Landscape structure: pollinators respond differently to urban and agricultural landscapes

Different methods are used to quantify urban and agricultural intensity. The most common method for estimating urban intensity is to quantify the amount of soil-sealing in the surrounding landscape (*Carlucci, Zambon & Salvati, 2019*; *Haase et al., 2012*; *Wellmann et al., 2018*). Soil-sealing is the outcome of covering soil with buildings, paving and roadways (*Concepción et al., 2016*). It presents barriers to mobility (landscape permeability) and fragments floral stands into discrete patches. Furthermore, it removes nesting sites for ground-nesting species (*Simao, Matthijs & Perfecto, 2018*; *Wojcik & McBride, 2012*). In contrast, many agricultural landscapes are characterized by large stands of monoculture crops. The large-scale use of pesticides and fertilizers present a different suite of barriers and stressors which fragment floral stands and limit landscape permeability (*Lenhardt et al., 2013*).

Studies which have compared the pollinator biodiversity between agriculture and urban landscapes have found them to rank similarly in diversity and richness but to contain different guilds in discrete and distinct communities (*Harrison, Gibbs & Winfree, 2019*; *Sattler et al., 2011*; *Wenzel et al., 2020*). Similarly, natural landscapes are not necessarily more biodiverse than agricultural and urban landscapes (*Collado, Sol & Bartomeus, 2019*), but have been shown to harbour a greater number of rare species (*Harrison, Gibbs & Winfree, 2019*). While the importance of natural landscapes for conservation cannot be
overstated (*Collado, Sol & Bartomeus, 2019*), protection of natural landscapes should be viewed as one of a suite of opportunities for providing suitable pollinator habitats, and the mitigation and restoration opportunities in non-natural landscapes need to be utilized to counter overall losses.

Both agricultural and urban landscapes can disproportionately benefit discrete groups or guilds of pollinators by providing resources favoured by those guilds (*Stein et al., 2018*). Urban habitats typically favour large-bodied insects which can range further between foraging resources (*Merckx, Kaiser & Van Dyck, 2018*) and cavity-nesting guilds, which take up residence in fences, walls and roof eaves (*Cane et al., 2006*; *Neame, Griswold & Elle, 2013*). Body size is also correlated with distance to fragments of natural habitat in agricultural landscapes (*Geslin et al., 2016*). This filtering occurs because larger-bodied insects can range further to find suitable food resources than smaller-bodied insects (*Geslin et al., 2016*). Furthermore, generalists are more likely to take advantage of crop species in agricultural landscapes (*Hopfenmüller, Holzschuh & Steffan-Dewenter, 2020*; *Leong, Kremen & Roderick, 2014*). In their meta-analysis of studies conducted globally on crop-visiting pollinators, *Kleijn et al. (2016)* found that the species pool in agricultural systems is made-up of a small subset of dominant species. Furthermore, dominant crop pollinators persist under agricultural expansion and can easily be enhanced by simple conservation measures, but threatened species are rarely observed on crops (*Kleijn et al., 2016*). Similar patterns of generalization have been observed in urban landscapes (*Leong, Kremen & Roderick, 2014*; *Lowenstein, Matteson & Minor, 2019*), however urban landscapes also act as a refuge for certain guilds of pollinators (*Wenzel et al., 2020*). Thus, they can provide supplementary seasonal foraging resources to wild and managed populations (*Koyama et al., 2018*).

Spill-over effects occur along the edges of habitat boundaries where species associated with one habitat type range into another from the boundary. They have been observed in several studies in croplands in which wild bee and non-bee pollinators spill-over from natural fragments, where they are more readily able to find suitable nest sites and a wide range of flower resources (*Ricketts et al., 2008*; *Simba et al., 2018*; *van Schalkwyk et al., 2020*). *Ricketts et al. (2008)* synthesized the results of 23 studies to determine the landscape effects on crop pollination services and found strong exponential declines in both pollinator richness and wild visitation rates with distance from natural and semi-natural vegetation (*Ricketts et al., 2008*). More specifically, in South Africa, flying insect abundance decreases in mango orchards with increasing distance to natural fragments (*Geslin et al., 2016*) and the proximity of natural patches to mango orchards benefits mango plantations both during flowering and when mangoes are not in flower. This is due to greater plant diversity in natural patches providing flowers over a protracted period (*Simba et al., 2018*). These phenomena motivate the inclusion of natural corridors and patches between agricultural crop fields, especially when animal pollination is required by the crop (*Carvalheiro et al., 2011*; *Geslin et al., 2016*; *Ricketts et al., 2008*; *Simba et al., 2018*).

Spill-over effects have also been observed from urban domestic gardens into adjacent agricultural areas, where insects which find suitable nesting sites in gardens are able to

supplement their diets from agricultural crops (*Langellotto et al., 2018*). This points to a complex pattern of exchanges in heterogeneous landscapes and indicates that urban landscapes are not necessarily as hostile as they appear at first glance (*Baldock et al., 2015*; *Groffman et al., 2014*; *Theodorou et al., 2020*; *Wenzel et al., 2020*).

Studies describing spill-over effects into urban landscapes have had mixed results, as some species increase with urbanization, some decrease, and others have no response (*Wenzel et al., 2020*). Urban landscapes are a complex patchwork of uses and local habitats, which do not always result in clear declines in abundance and richness as distance to natural or agricultural landscapes increases (*Theodorou et al., 2017*). This is because other environmental gradients within the city do not necessarily follow the same spatial patterns as soil-sealing. For example, changes in biodiversity have been documented along socio-ecological gradients with greater biodiversity recorded at the wealthier end of the income gradient (*Lubbe, Siebert & Cilliers, 2010*; *Qureshi, Haase & Coles, 2014*; *Shackleton et al., 2020*; *van Heezik et al., 2013*). Furthermore, residential gardens, and community lots offer both nesting sites and abundant floral resources throughout the year (*Glaum et al., 2017*; *Threlfall et al., 2015*). As such, they can provide refuge for certain guilds of pollinators (*Wenzel et al., 2020*) and supplementary seasonal foraging resources year-round (*Glaum et al., 2017*; *Koyama et al., 2018*).

The negative impacts of both soil-sealing and monoculture landscapes can be mitigated in several ways. Firstly, corridors and regular patches of natural vegetation can facilitate movement through the landscape by providing nesting and supplementary foraging resources (*Cane, 2015*; *Pauw & Louw, 2012*; *Simao, Matthijs & Perfecto, 2018*; *Wojcik & McBride, 2012*). Secondly, linear elements such as hedgerows and flower patches arranged in a linear pattern through the landscape further assist pollinators in orientating flight paths between larger natural or semi-natural patches (*Cranmer, McCollin & Ollerton, 2012*), thus preserving and establishing ecological stepping-stones and corridors (*Cranmer, McCollin & Ollerton, 2012*; *Knight et al., 2019*; *Ossola et al., 2019*; *Pauw & Louw, 2012*; *Riitters et al., 1995*; *Vrdoljak, Samways & Simaika, 2016*). Thirdly, where agricultural studies clearly articulate the value of natural remnants, urban studies have demonstrated that semi-natural, and artificially landscaped areas can provide ecologically rich surrogates and refuges in urban landscapes (*Anderson, Avlonitis & Ernstson, 2014*; *Dewaelheyns, Kerselaers & Rogge, 2016*; *Zhao, Sander & Hendrix, 2019*).

In many instances, the local land use and floral abundance have a greater influence on pollinator abundance and richness than landscape structure (*Simao, Matthijs & Perfecto, 2018*; *Theodorou et al., 2017*; *Threlfall et al., 2015*). MABES articulates this by placing the local site directionally ahead of the landscape structure, so that land use and management, and disturbance cycles are what determine the broader landscape structure and not the other way around (*Kremen et al., 2007*). This is noteworthy because it highlights the opportunities (and challenges) presented by the accumulation of small-scale interventions across a wider landscape (*Davis et al., 2017*; *Dewaelheyns, Kerselaers & Rogge, 2016*).

## Pollinator community structure: the importance of wild bees and non-bee pollinators in different landscape systems

Non-bees and wild bee species account for 39% of global pollination services (*Rader et al., 2016*), and several agricultural crops depend solely or heavily on pollination services from these taxa (*Gemmill-Herren & Ochieng, 2008*; *Martins & Johnson, 2009*; *Rader et al., 2016*). This number excludes informal reliance on wild pollinators by the urban poor. Despite this, studies on both urban (*Wenzel et al., 2020*) and agricultural systems (*Rader et al., 2016*) have overwhelmingly focused on bees. Although urban studies generally include wild bees as well as (managed) *Apis mellifera*, there is a paucity of information on non-bee taxa (*Rader et al., 2016*; *Wenzel et al., 2020*).

One area of research in South Africa which regularly considers wild pollinators, is for crops which contribute to the biodiversity economy. In 2013, the total value added by bio-products to the domestic retail market was R1.5 billion (*Republic of South Africa DEA, 2016*). The two largest resources grown from indigenous plants, *Aloe ferox* (bitter aloe) and *Aspalanthus linearis* (rooibos tea), are pollinated by birds in the case of the aloe (*Botes, Johnson & Cowling, 2009*) and indigenous wasps in the case of rooibos (*Gess, 2000*; *Gess & Gess, 2010*). Although this is indicative of the importance of wild Hymenoptera and non-bee pollinators to local agricultural production, the importance is not limited to indigenous crops. The carpenter bee *Xylocopa caffra* and *Macronomia rufipes* are both important pollinators for eggplant crops in Kenya (*Gemmill-Herren & Ochieng, 2008*). Papaya relies exclusively on pollination by hawkmoths (*Martins & Johnson, 2009*) and blowflies are an effective alternative to bee pollination in mango plantations (*Saeed et al., 2016*). Syrphid flies are among the most important pollinators, because they visit at least 72% of global food crops (*Doyle et al., 2020*; *Raguso, 2020*). In addition to floral resources, each of these taxa has differing brood and nesting requirements (*Gess & Gess, 2010*). Determining the nesting and habitat needs of non-bee species is therefore an important area for development in pollinator research, both for agriculture and urban landscapes.

Wild-bee and non-bee pollinators are not only important to agricultural production, but more specifically to the conservation and the persistence of natural plant communities, especially plants which are specialists and rely on the services of a limited number or single species of insect. Notably, South Africa has high degrees of endemism and several specialized pollinator systems (*Johnson & Steiner, 2003*; *Johnson & Wester, 2017*). Diversity of pollination systems in part explains the huge diversity of Iridaceae in the region (*Goldblatt & Manning, 2006*).

Two examples of non-bee pollinators which warrant special attention include Diptera and Hopliini. Long-proboscid fly pollination in southern Africa includes four separate syndromes using different sets of flies and plant species in different parts of the subcontinent (*Goldblatt & Manning, 2006*). Internationally, there are few urban studies which have included Diptera in the analysis of pollinator responses to urban form. In Italy, Diptera visitation to wildflowers in urban systems is independent of urban intensity (*Basteri & Benvenuti, 2010*), but hoverflies are negatively affected by urban intensity in the UK and Sweden (*Persson et al., 2020*; *Salisbury et al., 2015*). Thus, there is inadequate

evidence to be able to draw generalizations on Diptera responses to urban environments; this represents a gap in knowledge, despite their importance to pollination both in South Africa and globally (*Barraclough & Slotow, 2010*; *Doyle et al., 2020*; *Raguso, 2020*; *Saeed et al., 2016*).

Approximately 63% of the world's Hopliini (monkey beetles) species and 38% of genera are found in South Africa and over 50% of those are concentrated in the Cape Region, which is the global center for monkey beetle radiation (*Colville, Picker & Cowling, 2018*). No urban studies have been published on monkey beetles. Yet, they are the most abundant pollinators of Asteraceae and Aizoaceae in the succulent Karoo (*Mayer, Soka & Picker, 2006*) and are the primary pollinators for two genera of Hyacinthaceae, seven Iridaceae, one Hypoxidaceae, two Asteraceae, two Campanulaceae, and one Droseraceae (*Goldblatt, Bernhardt & Manning, 2013*). Studies on agricultural landscapes demonstrate beetle assemblage adjusts with flower community and disturbance patterns according to feeding guilds (*Colville, Picker & Cowling, 2002*; *Mayer, Soka & Picker, 2006*). A similar study to establish monkey beetle distributions in urban contexts would provide insights into the conservation potential of Asteraceae, Aizoaceae, and beetle-pollinated geophytes in urban landscapes.

## Local habitat: opportunities to improve connectivity and quality for urban pollinators

Local habitats do not contribute equally to pollination ecosystem service because different land-uses provide different levels of pollen resources (*Davis et al., 2017*; *Zhao, Sander & Hendrix, 2019*). *Zhao, Sander & Hendrix (2019)* determined whether pollinator supply in urban landscapes could meet demand for pollination services from urban agriculture. Using Iowa City, USA, as a case study, they established that pollinator surpluses occur in natural areas and heavily-vegetated residential neighbourhoods, and deficits occur in resource-poor lawns (*Zhao, Sander & Hendrix, 2019*). However, converting turf-grass to flower-rich gardens throughout the city would support increased supply of pollination services (*Davis et al., 2017*). To a lesser extent, this has been assessed in South Africa where civic-led groups restored urban sites (*Anderson, Avlonitis & Ernstson, 2014*). The restored sites measured insect diversity comparable to adjacent conservation areas despite differing floral communities (*Anderson, Avlonitis & Ernstson, 2014*). These studies point to the potential that even partially restored patches can generate ecological stepping-stones and refuges in the urban landscape for insect flower visitors (anthophiles), but also to the value that local-habitat level interventions can contribute towards mitigating the negative effects of urbanization.

Acknowledging that the amount of concrete in the surrounding landscape has a predominantly negative impact on most insect species at the most urbanized end of the urban gradient (*Wenzel et al., 2020*), studies which included multi-variate analysis of local habitats and landscape level patterns often found that the main drivers of urban pollinator communities were factors at local habitat scale such as floral abundance (*Bennett & Lovell, 2019*; *Davis et al., 2017*; *Simao, Matthijs & Perfecto, 2018*; *Wojcik &*

*McBride, 2012*), indicating that there are opportunities to supplement habitats at the local habitat scales.

Firstly, floral abundance plays a major role in pollinator assemblage. *Avuletey & Niba (2014)* found stronger influences of local habitat quality in Mthata city, South Africa. They compared butterfly assemblage across sampling units from a nature reserve to an urban center. Highly disturbed sampling units with fewer nectaring plant species attracted fewer butterfly species, but higher richness and abundance of butterflies were recorded outside of the reserve when compared with sampling units inside the reserve. They recommended improving the micro-habitat conditions in the nature reserve and the establishment of corridors into the city in order to attract the butterflies to the reserve (*Avuletey & Niba, 2014*). Further afield, in New York, sunlight and floral abundance are the major factors predicting local bee and butterfly diversity in densely populated neighborhoods (*Matteson & Langellotto, 2010*). *Wojcik & McBride (2012)* sum up the implications of these studies: "management strategies that provide dense and abundant floral resources should be successful in attracting (pollinators), irrespective of their location within the urban matrix" (*Wojcik & McBride, 2012*, p. 581). Thus, the establishment of floral patches in the most urbanized parts of the city provides habitat resources for urban pollinators, especially if they are arranged to facilitate movement between larger floral-rich patches (*Cranmer, McCollin & Ollerton, 2012*).

Secondly, patches of floral resources can be successful in supporting pollinator mobility even at very small scales. *Simao, Matthijs & Perfecto (2018)* experimentally introduced potted *Lobularia maritima* (sweet alyssum) and monitored halictid visitation. They found that smaller flower plantings may have higher impacts on small pollinators than larger plantings, and suggested that smaller floral plantings across the landscape may provide a niche for smaller bees (*Simao, Matthijs & Perfecto, 2018*). This idea is reflected by *Plascencia & Philpott (2017)*, who found that large stands had less richness than patchy floral stands due to competition from managed hives at the more extensive sites (*Plascencia & Philpott, 2017*). This lends itself to a conservation strategy which encourages citizens to adopt creative solutions such as butterfly balconies or planted roofs. Caution is offered, however, in the discussion of planted roof gardens because they favor thermophilic species, and do not support succession process (*Mayer, Soka & Picker, 2006*), and unless they provide a diversity of floral resources, the contribution to providing pollinator habitat performs no better than turf-grass (*Kratschmer, Kriechbaum & Pachinger, 2018*; *Ksiazek Mikenas et al., 2018*; *Tonietto et al., 2011*; *Williams, Lundholm & Scott MacIvor, 2014*).

## Flower community structure: effects on pollinator diversity

In agricultural landscapes, studies which focused on supplementary floral resources emphasised promoting the health of managed honeybee hives by investigating seasonal food sources when hives are not being deployed to pollinate croplands (*Koyama et al., 2018*; *Langellotto et al., 2018*; *Melin et al., 2014, 2020*). In South Africa, two sub-species of *Apis mellifera* (*A. mellifera capensis* and *A. mellifera scuttelata*) occur naturally in the wild. Regionally, managed hives provide 50–98% of animal driven pollination services in

croplands (*Melin et al., 2014*; *Rader et al., 2016*) and *A. mellifera* spp. are the dominant pollinators for most of the crops requiring animal-facilitated pollination (*Melin et al., 2014*). Peak requirements for pollination occur with the mass flowering of crops which are limited to specific times of the year (spring and autumn). In order for managed honeybee hives to remain "strong" throughout the year, they are rotated between crops and "off-season" resources. As such, when the crop season ends, commercial beekeepers routinely move hives to stands of *Eucalyptus* or natural vegetation so that they have access to food when not in use for crop pollination (*Johannsmeier, 2016*; *Masehela, 2017*; *Melin et al., 2014*). *Eucalyptus* is preferred by *Apis mellifera* as a resource in mixed landscapes, accounting for 41% of the pollen loads collected (*Melin et al., 2020*). However, an increase in the area of *Eucalyptus* stands results in an increase in *Eucalyptus* as well as wild pollen collection but a reduction in the proportion of exotic pollen collection (*Melin et al., 2020*), suggesting that exotic plants are the least-favoured of the three taxa and pointing to other ways in which mass-flowering plants compete or facilitate pollination of their neighbours (*Schmid et al., 2016*). Some headway has been made into testing other mass-flowering species. In particular, *Acacia* spp.—most notably *Acacia senegal* is identified by *Martins (2004)* for playing a similar role in building queen strength and generating honey stores in managed hives in central Africa. The majority of the pollen produced by the floral display is excessive and ends up distributed on the ground (*Martins, 2004*). The natural distribution of *A. senegal* stretches from the Arid parts of central Africa to the northern borders of South Africa, however assessments of bee pollen potential ranked it best among the other native *Acacia* spp. (*Johannsmeier, 2016*). Despite this ranking, the listing of "lesser" indigenous *Acacia* spp. were delivered with considerably more uncertainty, highlighting the need for further investigation (*Johannsmeier, 2016*; *Masehela, 2017*).

That different species of flowers are preferentially utilized by anthophiles has been investigated across several urban studies (*Baldock et al., 2015*; *Garbuzov, Alton & Ratnieks, 2017*; *Garbuzov & Ratnieks, 2014*, *2015*; *Lowenstein, Matteson & Minor, 2019*; *Martins, Gonzalez & Lechowicz, 2017*; *Michołap, Kelm & Sikora, 2018*; *Yang et al., 2019*). In North America, one third of urban flowering plants are not visited by pollinators, and of those that are, 40% are visited more frequently than others (*Lowenstein, Matteson & Minor, 2019*). When urban foraging patterns are compared with natural and agricultural landscapes, urban pollinators forage from a smaller proportion of available plant species (*Garbuzov, Alton & Ratnieks, 2017*; *Martins, Gonzalez & Lechowicz, 2017*; *Baldock et al., 2015*). This is likely due to the high prevalence of introduced and exotic species in urban landscapes. For the same reason urban landscapes have recorded higher visitation rates of individual plants, but lower rates of seed, with the net result of reduced pollination services to all plants (*Leong, Kremen & Roderick, 2014*). In other words, gardeners are selecting highly diverse plant species for their gardens at the expense of the genetic pool in the local population. Too few individuals from too many different species are planted together to provide enough genetic material and pollination efficiency for strong plant metapopulations and reproductive success.

This can be explained by market forces which play out in the supply and demand of plants sold at garden centres and misinformation within the public domain (*Garbuzov, Alton & Ratnieks, 2017*). Gardeners seeking to plant flowers to encourage and provide resources for pollinators are often given erroneous information, such that plants unattractive to wild pollinators are recommended by garden center staff and marketing material, and those that are attractive are overlooked (*Garbuzov & Ratnieks, 2014*). To address this phenomenon, several international studies have identified subsets of flowers that are classified as "highly" attractive to pollinators (*Garbuzov, Alton & Ratnieks, 2017*). These lists provide scientifically verified recommendations of flower resources for a city or region (*Garbuzov, Alton & Ratnieks, 2017*; *Garbuzov & Ratnieks, 2014*, *2015*; *Michołap, Kelm & Sikora, 2018*; *Yang et al., 2019*). Similar studies in the South African context are rare where the focus is on rather to study mutualisms, pollination syndromes, and flower resource use in natural landscapes (*Goldblatt & Manning, 2006*; *Johnson & Midgley, 2001*; *Johnson & Steiner, 2003*; *Johnson & Wester, 2017*; *Raguso, 2020*). Regional information is therefore available but has not been tested for preference and pollination loads in the way *Eucalyptus* has, nor has it been collated for the purposes of informing gardeners, garden centres and staff. This represents an important gap in knowledge about the relative contribution which flowering plants can make to boosting pollinator populations both within the urban landscape and as a potential alternative to *Eucalyptus* for managed hives.

Further to this, the potential contribution of "weedy" flowers and ruderal species should not be overlooked (*Carvalheiro et al., 2011*; *Colville, Picker & Cowling, 2002*; *Koyama et al., 2018*; *Lowenstein, Matteson & Minor, 2019*; *Martins, Gonzalez & Lechowicz, 2017*; *Wilson & Jamieson, 2019*). In sunflower fields in Limpopo, weed diversity increases anthophile diversity, thereby ameliorating the effects of distance to natural vegetation patches and increasing pollination success and seed set (*Carvalheiro et al., 2011*). In urban landscapes, the contribution of "weedy" and exotic species to pollination resources ranks below perennial and indigenous plants but there are examples where individual species of exotic weeds receive disproportionate visitation rates in turf-grass and other urban landscapes (*Larson, Kesheimer & Potter, 2014*; *Lerman et al., 2018*; *Lowenstein, Matteson & Minor, 2019*). On an urban to rural gradient, *Stenchly et al. (2017)* noted a turnover in the weed-species present in Okra field, from those which rely on free-roaming animals and livestock for dispersion in rural and peri-urban environments, to those which rely on birds for dispersion in urbanized areas (*Stenchly et al., 2017*).

Proponents of the contribution that common "weeds" can make, advocate for reduced mowing frequency and lazy weeding of turf-grass as a landscape management strategy for supporting urban pollinators (*Bertoncini et al., 2012*; *O'Sullivan et al., 2017*; *Russell et al., 2018*; *Yang et al., 2019*).

## Humans as facilitation agents

Whether or not urban landscapes can provide more connected landscape habitats for the support of healthy pollinator populations will depend on the implementation of

pollinator-friendly landscape designs and maintenance practices (*Kremen et al., 2007*; *Senapathi et al., 2017*). *Kremen et al. (2007)* conceptualize this in their MABES model as the valuation of the desired eco-system service as an informant of policy; however, for public open space and domestic yards different mechanisms are likely to be at play in informing design and management outcomes. Specifically, it depends on whether there is citizen buy-in for the implementation of landscaping practices which positively drive pollinator populations. In an agricultural landscape or a hybrid, urban agriculture landscape, pollination service delivery likely only competes with pest management and the trade-off between available land and crop carrying capacity. Ultimately the goal of increased yields and increased pollination services are aligned (*Tamburini et al., 2019*).

In urban landscapes, public parks, road verges, vacant plots, residential suburban gardens, and community gardens provide various ecosystem services including cultural, social, recreational, provisioning, and regulating services (*Bolund & Hunhammar, 1999*). Some ecosystem services co-exist and overlap in space and time, whilst others are mutually exclusive in the apportionment of space. The provision of pollinator habitats will therefore have to compete against other community needs for dedicated space. Furthermore, whether or not a park manager or landscaping contractor is able to implement the planting of mass-flowering species, or reduced mowing frequency, depends on the community tolerance of the related ecosystem disservices, such as airborne pollen and long grass (*Kremen et al., 2007*).

Community parks are limited in space (especially at the low end of the socio-economic gradient) (*Shackleton et al., 2020*) and should be designed and managed with community aspirations for exercise, aesthetics, and social gathering in mind (*Bolund & Hunhammar, 1999*). Pollinator gardens are not necessarily in conflict with these uses, but where space is limited, there could be trade-offs and other ecosystem services will need to be considered. Citizen engagement is therefore essential when proposing landscaping changes. In this respect, a "learning by doing" approach may be the most appropriate way to implement urban planning and design. This would involve a process of engaging public participation to collaboratively value or rank the public goods and ecosystem services provided by public open space (*Ahern, Cilliers & Niemelä, 2014*; *Cilliers & Siebert, 2012*), and to develop an integrated plan for the management of parks and gardens. Successful examples of civic stewardship action already exist in South African cities (*Anderson, Avlonitis & Ernston, 2014*; *Cilliers & Siebert, 2012*), implying that there are opportunities to engage the public in the production of pollinator habitats.

The discussion above has highlighted several recommended interventions for improving pollinator habitats in urban cities. Ultimately these are design and landscape management outcomes. Table 2 summarizes the opportunities according to the components in the MABES model and provides the references where they are discussed in more detail.

**Table 2 Summary of opportunities for supporting pollinators in urban habitats.**

| MABES component | Opportunity/recommendation | References |
|---|---|---|
| Local Habitat | Small patches of abundant flowers interspersed throughout the landscape | *Plascencia & Philpott (2017), Simao, Matthijs & Perfecto (2018)* |
| Flower community | Targeted species/pollinator syndromes to support mutualisms (*e.g.* red and orange tubular flower for sunbirds) | *Pauw & Louw (2012)* |
| Local Habitat | Increased flower abundance | *Bennett & Lovell (2019), Theodorou et al. (2020), Vrdoljak, Samways & Simaika (2016)* |
| Local Habitat | Semi-natural patch rehabilitation through citizen action | *Anderson, Avlonitis & Ernstson (2014)* |
| Flower community | Produce attractive species lists of flowers for dissemination to the public | *Garbuzov & Ratnieks (2014), Michołap, Kelm & Sikora (2018)* |
| Flower community | Altered mowing regimes to allow for lawn flowers to blossom | *Bertoncini et al. (2012), Knight et al. (2019), Lerman et al. (2018), O'Sullivan et al. (2017), Yang et al. (2019)* |
| Flower community | Replace turf grass with floral rich landscapes | *Davis et al. (2017), Zhao, Sander & Hendrix (2019)* |
| Landscape structure | Corridors and linear elements | *Avuletey & Niba (2014), Cane et al. (2006), Cranmer, McCollin & Ollerton (2012), Wojcik & McBride (2012), Knight et al. (2019)* |
| Landscape structure | Minimize impervious cover | *Bennett & Lovell (2019)* |
| Pollinator community | Establish the value of wild and non-bee pollinators and develop urban strategies to cater for the unique needs of different taxa | *Botes, Johnson & Cowling (2009), Gemmill-Herren & Ochieng (2008), Gess (2000), Martins & Johnson (2009), Johnson & Steiner (2003)* |

## CONCLUSIONS

This paper set out to review the South African and international literature to determine the potential that urban environments have to provide refuge and supplementary foraging resources to both wild and managed pollinators. We compared and contrasted findings in an urban setting from international studies with those emerging from the agricultural landscapes in South Africa, noting the structural differences between agricultural and urban land-use and management.

While evidence from international studies suggest that certain pollinator guilds can do relatively well in urban landscapes with potential for spill-over effects from urban habitats to adjacent agricultural croplands or to provide supplementary nesting and foraging habitats to urban agriculture, the vast majority of studies reported mixed-results, resulting either from factors other than urban intensity such as local floral abundance, landscape connectivity, or tree canopy, or due the study focusing on broad taxonomic mix. Many studies reported that local habitat structure played a more important role in pollinator health than did the level of urban intensity in the surrounding landscape.

In South Africa, communities at the lower end of the socio-economic gradient in the peri-urban periphery, often rely on both urban agriculture and wild foraged provisions to supplement their livelihoods, thereby placing the importance of pollination services for these communities as paramount. Several management practices can therefore be employed to boost the quality of foraging and ease of mobility through the city at different scales. At the landscape scale, providing corridors and natural patches will provide refuges for urban pollinators. But within the urban fabric at the local habitat scale, there are

strategies which can mitigate the hostility of between-patch landscapes. These include replacement of turf grass with flower rich plants, strategic grass-cutting practices to allow for "weedy" plants to grow, rehabilitation of areas adjacent to managed conservation areas, and small-scale plantings in the most urbanized parts of the city.

Several gaps and future directions for research were highlighted. The potential of South African cities to provide habitat for wild pollinator conservation and supplementary floral resources for managed honeybee hives remains unexplored to date. In South Africa, there are very few urban pollinator studies and therefore the potential they hold to support wild bee and non-bee pollinators is relatively unknown. Specific attention should be given to closing the gap in knowledge on Halictids, Diptera, Lepidoptera, and Hopliini, both locally and internationally. In South Africa, there is a gap in knowledge of relative flower attractiveness to wild and managed pollinators and a need to verify the information being disseminated in garden centers and the public domain. Better understanding of the potential from individual indigenous plants to provide pollen at levels similar to *Eucalyptus* is needed to determine if indigenous flowering trees can be used to supplement managed hives and wild-pollinator diets. Specifically, there is a need to establish greater confidence in the level/classification of pollinator resources provided by the multitude of native Acacia species for supplementary use by beekeepers.

## ACKNOWLEDGEMENTS

We would like to acknowledge Johnathan Colville for his advice and feedback on early drafts of this paper.

### Funding

Peta Brom was supported by the SASAC/NRF Scholarship (SASAC/170914262644). The funders had no role in study design, data collection and analysis, decision to publish, or preparation of the manuscript.

### Grant Disclosures

The following grant information was disclosed by the authors:
SASAC/NRF Scholarship: SASAC/170914262644.

### Competing Interests

The authors declare that they have no competing interests.

### Author Contributions

- Peta Brom conceived and designed the experiments, performed the experiments, analyzed the data, prepared figures and/or tables, authored or reviewed drafts of the paper, and approved the final draft.
- Les G. Underhill conceived and designed the experiments, analyzed the data, authored or reviewed drafts of the paper, and approved the final draft.
- Kevin Winter conceived and designed the experiments, authored or reviewed drafts of the paper, and approved the final draft.

## Data Availability

The raw data is available at the Supplemental File.

## Supplemental Information

Supplemental information for this article can be found online at http://dx.doi.org/10.7717/peerj.12788#supplemental-information.

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
