# Peer review of "A review of the opportunities to support pollinator populations in South African cities"

_PeerJ, doi:10.7717/peerj.12788_

## Round 0.1 · original submission · Major Revisions

· Academic Editor

Major Revisions

Thank you for your submission. Based on the reviewer comments this paper requires major revisions. I look forward to receiving the edited document.

Reviewer 2 has suggested that you cite specific references. You are welcome to add it/them if you believe they are relevant. However, you are not required to include these citations, and if you do not include them, this will not influence my decision.

Reviewer 1 ·

Basic reporting

I think the literature is nearly sufficiently reflected, in light of being South African specific I find the literature review in respect to the data available from South Africa rather weak and lacking depth.
The authors data is shared but it is not really clear to me why they did the literature search if they do not use the results thereof.

Experimental design

It brings me to the point that I am not convinced that they showed that one can use the data from the global north on urban data and compare that to the agricultural setup in South Africa and then conclude onto the South African urban set up.

Validity of the findings

It is one of the first reviews putting it into a Southern African context, so I am not disagreeing with the conclusions but the way how they get to them should be more detailed and focused.

Additional comments

Dear authors,

I find the review well written, however, the title is misleading since I seems to have missed the opportunities for South Africa. Moreover, the link / transition of agriculture vs urban land-use to being able to add the international studies is rather weak in my opinion and should be one of the more detailed aspects, convincingly showing that one can interchange agri and urban, especially looking at line 505 “… structural differences between agriculture and urban…”

Since the review wants to have a more South African perspective I am surprised that the authors only relied on published papers from South Africa and not included thesis work or conference proceedings. I am sure Anderson is doing work on that topic for a while and there is more data available.

I suggest that either broaden it and delete “South African” from the title, or find more South African data and show that one can actually make conclusion based on agricultural studies to an urban setting.
Another option would be by expanding it to “African” cities and include the data from Africa, rather than South Africa.

What about the Ruan Veldtman’s work and the FAO project - I miss some of their publications

Why limited the agricultural aspect to indigenous plants - what about citrus industry?
Why only Eucalyptus, that seems only important in the Western Cape. What about the rest of the country.
Why did you actually do the literature search if you do not really use it and analysis it. Table 1 is a selective summary of the literature

Also, I miss the importance of Apis for Fynbos
Several places were Apis mellifera is misspelled

Line 276 could have more South African Examples see below


A summary graph would be helpful to better capture the results in the supp


Spelling line 181,

Archer, C.R., Pirk, C.W.W., Carvalheiro, L.G., and Nicolson, S.W. (2014). Economic and ecological implications of geographic bias in pollinator ecology in the light of pollinator declines. OIKOS 123, 401-407.10.1111/j.1600-0706.2013.00949.x
Carvalheiro, L.G., Seymour, C.L., Veldtman, R., and Nicolson, S.W. (2010). Pollination services decline with distance from natural habitat even in biodiversity-rich areas. Journal of Applied Ecology 47, 810-820.10.1111/j.1365-2664.2010.01829.x
Carvalheiro, L.G., Veldtman, R., Shenkute, A.G., Tesfay, G.B., Pirk, C.W.W., Donaldson, J.S., and Nicolson, S.W. (2011). Natural and within-farmland biodiversity enhances crop productivity. Ecology Lett. 14, 251-259.10.1111/j.1461-0248.2010.01579.x

Reviewer 2 ·

Basic reporting

This is a well-written and detailed review around the topic of urban pollinators and their conservation, management and appreciation.

The review brings together global literature and connects the different aspects of the current state of the science with a more focused analysis of the context of urban areas in South Africa.

Experimental design

The study design is adequate.

A bit more effort could be made to access some of the literature and evidence about urban pollinator programs and conservation more widely in Africa. While some of this is in what would be considered 'grey literature' it provides an important context on what the current state of things are - which is what the review intends to do.

Validity of the findings

Overall the findings are valid and well-presented.

One aspect that is lacking is the bias towards both western urban areas and a lack of detail about the different urban cultures in South Africa. Given the history of South Africa, it would be prudent to present this context, as it has serious impacts for human settlements and thus where biodiversity is conserved, or not, within urban areas. For example, segregation of cities historically has major impacts for current patterns of green space and who can access them. While this is not a focus of the review, it does have very real impacts in the present day situation in urban areas.

Additional comments

This is a well-written and detailed review around the topic of urban pollinators and their conservation, management and appreciation.

The review brings together global literature and connects the different aspects of the current state of the science with a more focused analysis of the context of urban areas in South Africa.

A few minor points to consider:

There is a vast and rich literature on urban agriculture in Sub Saharan Africa and other developing countries. I suggest adding this as it provides a wider context and also noting that the perceived boundaries between urban-rural or protected-developed are blurry in many areas:

Lee-Smith, D., 2010. Cities feeding people: an update on urban agriculture in equatorial Africa. Environment and urbanization, 22(2), pp.483-499.

Foeken, D.W.J., Owuor, S., de Bruijn, M.E. and Dijk, R.A., 2001. Multi-spatial livelihoods in sub-Saharan Africa: rural farming by urban households-the case of Nakuru town, Kenya. African dynamics, pp.125-139.

Speybroeck, N., Berkvens, D., Mfoukou-Ntsakala, A., Aerts, M., Hens, N., Van Huylenbroeck, G. and Thys, E., 2004. Classification trees versus multinomial models in the analysis of urban farming systems in Central Africa. Agricultural Systems, 80(2), pp.133-149.

Across many parts of West and East Africa, there are increasing efforts to have public green spaces that are multi-functional and many local conservation or citizen organisations are engaged in developing these. While much of this is outside the core scientific literature, it would be worth a mention in the paper as it provides a more nuanced and comprehensive picture of what is currently happening:

https://friendsofcitypark.org/pollinator-garden-gets-a-jump-start-with-a-boscowen-makeover/

There are a number of more recent reviews that provide a wider context of pollinator conservation, management and the many complex issues that intersect with their wider understanding and appreciation. I suggest that these references be added to the the ms:

Hall, D.M. and Martins, D.J., 2020. Human dimensions of insect pollinator conservation. Current opinion in insect science, 38, pp.107-114.

Potts, Simon G., et al. The assessment report on pollinators, pollination and food production: summary for policymakers. Secretariat of the Intergovernmental Science-Policy Platform on Biodiversity and Ecosystem Services, 2016.

There is a growing appreceation of the potential impacts of more dense urban beekeeping efforts on wild or non-managed pollinator species in urban landscapes. Given the wider issue that there is still limited understanding of biodiversity and its’ connection with human life and livelihoods in many areas, this is subject that might be worth a bit more of a discussion in this review.

Reviewer 3 ·

Basic reporting

The present research reviews the international literature about opportunities to support pollinator population and compare with the few studies realized in South Africa in the agricultural landscape. With the lack of studies about this subject in South Africa, authors tried to do some parallels with the urban landscape.

While the main part of the article is very interesting to read with a good English used throughout, the introduction appears a little bit disconnected. The introduction is difficult to follow, and I suggest to the authors to maybe merge their actual introduction with the first part of their review entitled “A CONCEPTUAL MODEL FOR ANALYSING THE INTERACTIONS OF MOBILE AGENTS (POLLINATORS) » mainly based on the work of Kremen et al. 2007.

I have some concerns about the title and the lack of presentation or description of South African cities. Maybe it could be interesting to highlight some specificities of the South African context in introduction or develop more within each part of the review.
Regarding the introduction, I think it could contain more information about the decline of pollinators, why urban landscape could be favorable for some taxa. Please detail for which taxa it is favorable or not, and add some precision of the work of Wenzel 2020 which is described L51-54 (see miner comments). The third paragraph of the introduction is not easy to follow as a reader and I did not completely understand why there is a switch for urban to agricultural landscape. I understand that some factor could act similarly in urban and agricultural landscape but I think it is risky to do the parallel. It could be better to reformulate this paragraph.

I don’t see what is the input of the figure 1 which is the same figure as Kremen et al. 2007.

Experimental design

The survey methodology is well described. Maybe authors could provide the total number of papers they included in their review (158 papers).

The review is organized as follows :
1. A CONCEPTUAL MODEL FOR ANALYSING THE INTERACTIONS OF MOBILE AGENTS(POLLINATORS)
2. POLLINATOR COMMUNITY STRUCTURE: THE IMPORTANCE OF WILD BEES AND NON-BEE POLLINATORS IN DIFFERENT LANDSCAPE SYSTEMS.
2.1. LANDSCAPE STRUCTURE: POLLINATORS RESPOND DIFFERENTLY TO URBAN AND AGRICULTURAL LANDSCAPES
2.2. LOCAL HABITAT: OPPORTUNITIES TO IMPROVE CONNECTIVITY AND QUALITY FOR URBAN POLLINATORS
2.3. FLOWER COMMUNITY STRUCTURE: ITS EFFECTS ON POLLINATOR DIVERSITY
2.4. HUMANS AS FACILITATION AGENTS

I suggested merging the first part of the review with the introduction of the review as the authors were based only on the work of Kremen et al 2007.

Validity of the findings

The review highlight the lack of knowledge about the effect of urbanization on pollinators in South Africa. Authors only provide information in some bee taxa and insects taxa at the end of the manuscript whereas they could mentionned some studies in the main text.

Development and Future of Insect Conservation in South Africa https://link.springer.com/chapter/10.1007/978-94-007-2963-6_12
Lepidopterology in Southern Africa: Past, Present and Future
https://link.springer.com/chapter/10.1007/978-94-007-2963-6_12

Some studies on plant diversity across urban-rural gradient could also be mentionned in the FLOWER COMMUNITY STRUCTURE part.

Impacts of urbanization in a biodiversity hotspot: Conservation challenges in Metropolitan Cape Town https://www.sciencedirect.com/science/article/pii/S0254629910001390

Post-apartheid ecologies in the City of Cape Town: An examination of plant functional traits in relation to urban gradients
https://www.sciencedirect.com/science/article/abs/pii/S0169204619301197

Ecological outcomes of civic and expert-led urban greening projects using indigenous plant species in Cape Town, South Africa
https://www.sciencedirect.com/science/article/abs/pii/S0169204614000802

Even it is not the main subject here, maybe you can do some hypothesis about these previous works.

Additional comments

Minor comments :

Abstract

L21-22. What do you mean by regions and taxa ? This sentence is not clear for me.
L27. Which taxa ? Please precise and reformulate the sentence by avoiding “do well”.
L33-34. Why do you finish by some precise advice only for honeybees ? Please stay wide for the conclusion sentence.

Introduction

L38-41. Please merge the 2 sentences, I suggest, “In the context of global decline of insect species and biomass (references), there is growing evidence that …. (References).”
L43. Remove « indicating that”.
L51-54. How many studies have been referenced in the work of Wenzel 2020 in total ?
L63. Replace “this” by “there”.

Review parts
L181. Please use Apis mellifera instead Apis melifera all along the manuscript.
L390. Remove "for"
L396-397. Please reformulate.

---

## Round 0.2 · Minor Revisions

· Academic Editor

Minor Revisions

Hi there,

Thank you for addressing the reviewer comments in your resubmission. Please remove the dotted line in fig 1b and see the reviewer comments attached.

Reviewer 1 ·

Basic reporting

I suggest to remove the dotted line in figure 1b, since you did not report on any stats, nor any descriptive information (like slope) the line is not useful at all and even with stats & slope I am not sure that it would add value.

Experimental design

I think the authors addressed the critical points raised

Validity of the findings

The authors refocused the MS significantly and I think they make a convincing case for a South African focus

Additional comments

Without the line in figure 1b I would have said "accepted as is"

---

## Round 0.3 · accepted · Accept

· Academic Editor

Accept

Thank you for making the suggested revisions.